# Dynamic Heat Dissipation Model of Distributed Parameters for Oil-Directed and Air-Forced Traction Transformers and Its Experimental Validation

**DOI:** 10.3390/e25030457

**Published:** 2023-03-06

**Authors:** Yonghua You, Kun Shao, Zhengming Yi

**Affiliations:** 1State Key Lab. of Refractories and Metallurgy, Wuhan University of Science and Technology, Wuhan 430081, China; 2International Research Institute for Steel Technology, Wuhan University of Science and Technology, Wuhan 430081, China

**Keywords:** modeling, temperature rise experiment, dynamic heat dissipation, traction transformer

## Abstract

A traction transformer with narrow oil channels is usually cooled with the ODAF or “Oil Directed Air Forced” method, where its temperature greatly depends on the Joule heat of windings, the conjugate heat transfer in the transformer, and the secondary heat release via oil cooler, together with the oil flowrate generated by oil pump. Neither the thermal–electric analogy nor the CFD simulation approach is qualified to predict the temporal and spatial temperature variations in this type of transformer. In the current work, the distributed parameter models are built for traction transformers and oil coolers with the assumption of a one-dimensional temperature field in the oil flow direction, respectively. Then, the two models are combined with the lumped parameter ones of oil pumps and pipes via the flow rate, temperature and pressure continuities at their interfaces, resulting in the derivation of the dynamic heat dissipation model of oil-directed and air-forced traction transformers. Additionally, an efficient algorithm is proposed for its numerical solution, and the temperature rise experiment is performed for model validation. Finally, the fundamental of dynamic heat dissipation in traction transformers is investigated with the current numerical model and the effects of ambient temperature are studied.

## 1. Introduction

Transformers take their winding temperatures rising with time due to the Joule heat of copper wires, which can result in function failure and life expectancy decrement. Many methods were proposed and applied to promote the cooling of transformers [1]. Oil-immersed transformers have a better heat dissipation behavior compared with dry-type counterparts [2] and are qualified to run at a larger power. The ONAN or “Oil Natural Air Natural” method, where the oil circulating due to buoyancy absorbs the heat generation of windings and then releases it to the ambient via radiation and natural convection, has the advantage of compact structure and reliable operation; however, its heat dissipation capacity is limited. As is well known, the heat transfer rate can be enhanced by increasing fluid velocity with a fan or pump. Thus, the ONAF or “Oil Natural Air Forced” and OFAF or “Oil Forced Air Forced” schemes are applied to cool the transformers with a larger heat generation rate. On the other hand, various mathematic models were proposed to study the heat dissipation of transformers. In the IEEE Std C57.119 [3], the increment of hottest-spot winding over ambient temperature is composed of its hottest-spot rise over top-oil temperature and the top-oil rise over ambient temperature, and the two parts are assumed to take the exponential rise or fall variation trend. These equations for the temperature variations require the bottom-oil rise over ambient at the rated conditions. With the assumption of lumped temperatures, Swift et al. [4] set up the differential energy equations of equivalent circuits for the oil and windings in transformers and performed experimental work to validate their thermal transformer model. They found that the exponential response was not the best for the cooling dynamics of the transformer [5]. It is well known that the thermal model of lumped parameters cannot identify the local temperature variation in transformers. To solve the problem, different modeling strategies were proposed [6,7,8,9,10,11,12,13,14]. To list a few, Zhang and Li [6] developed a two-dimensional thermal model to study the hot-spot temperature field of winding disks in an oil-immersed transformer, where the differential heat conduction equation of windings with a non-uniform heat generation was coupled with the non-isothermal hydraulic model. Torriano et al. [7] adopted the CFD software ANSYS CFX to simulate the conjugate heat transfer in an individual conductor of the transformer, and the effects of inlet temperature profile and mass flow rate on the flow and temperature fields were studied. Skillen et al. [8] extended the above work by covering all the passes of windings. It was observed that hot plumes occurred in horizontal ducts due to an inhomogeneous temperature profile, and the hot fluid streaks convected from one pass to the next. Wakil et al. [9] conducted numerical simulations to research the effects of geometric configuration and oil flow rate on transformer heat dissipation. Some rolls were found at the top and bottom of the windings, which helped to generate a homogeneous temperature field. To reduce the computation load, Gastelurrutia et al. [10] built the Complete and Slice models for an oil-immersed transformer. The former reproduced all the important geometries, and the oil channels and coils were treated as a porous zone, while the Slice modeled the fluid in a vertical cut of transformer and the oil channels were treated in a realistic way.

Traction transformers of EMU (Electric Multiple Units) trains run with a large heat generation rate and their winding temperatures vary notably with time. The advanced cooling scheme of ODAF or “Oil Directed Air Forced”, which overtakes the OFAF due to the thermal augmentation of oil directed through narrow channels between adjacent windings, is widely used for the heat dissipation of ODAF transformers, where the temperature field depends on the winding Joule heat, the conjugate heat transfer in transformer and the secondary heat dissipation in the oil cooler, together with the oil flow rate. When the CFD approach is adopted, the computation cell number is quite large because both the transformer and oil cooler need be modeled. Additionally, the oil flow rate through the ODAF transformer can be hard to determine in the CFD model. On the other hand, the lumped parameter model of the transformer based on the thermal–electric analogy cannot capture the local temperature field. As far as authors know, the literature on the flexible thermal model of ODAF transformers is limited.

In the current work, the distributed parameter models are set up for traction transformers and oil coolers, respectively. Then, the two models are combined with the lumped parameter ones of pump and pipes via the flow rate, temperature and pressure continuities at interfaces, and thus the mathematical model for the heat dissipation of traction transformers is derived. Next, an efficient numerical solution algorithm is proposed, and a temperature rise experiment with a traction transformer is performed for model validation. Finally, the dynamic heat dissipation process in traction transformers is studied with the current model.

## 2. Mathematical Model and Numerical Solution

### 2.1. Mathematical Model

Figure 1 depicts the heat dissipation process of traction transformer with the ODAF scheme. The transformer has two coil columns of rectangular conductors. The windings are wrapped on the round core limbs with the layered feature, which results in many narrow oil channels between adjacent layers of low- and high-voltage windings, as depicted in the A-A view of Figure 1. During the operation, the cool oil from oil coolers flows through the channels and then mixes at the outlet of the transformer, while the hot oil, having absorbed some Joule heat of windings, releases its thermal energy to the ambient via two oil coolers. The oil coolers feature the plate-fin type, and the ambient air is forced by a fan to flow through the air-side channels. The oil flows with a turn in the oil cooler. With the consideration that the air-side inlet temperature is the same for the two oil passes, the oil cooler can be treated as a one-pass heat exchanger with the oil-side length doubled.

The oil channels between the adjacent layers of low- or high-voltage windings take the same width, and the flow and heat transfer are assumed to be the same there. Additionally, the thickness of single-layer winding is small, and the copper conductor has a large heat conductivity; thus, the winding temperature is quite uniform in the cross-section. With the above considerations, the following one-dimensional version of dual-energy equations is used to express the heat transfer of oil and windings in traction transformers [15].
(1)$$\rho_{o}c_{p,o}\frac{\partial T_{t,o}}{\partial t}+\rho_{o}c_{p,o}u_{t}\frac{\partial T_{t,o}}{\partial x}=\lambda_{o}\frac{\partial^{2}T_{t,o}}{\partial x^{2}}+h_{t}a_{t}\frac{\alpha_{t,w}}{\alpha_{t,o}}\cdot \Delta T_{wo}$$
(2)$$\rho_{w}c_{p,w}\frac{\partial T_{t,w}}{\partial t}=\lambda_{w}\frac{\partial^{2}T_{t,w}}{\partial x^{2}}-h_{t}a_{t}\Delta T_{wo}+S_{JH}$$
where *u* is fluid velocity and depends on the actual oil flowrate of the oil pump, while *T*, *t* and *x* are temperature, time and streamwise displacement, respectively; *ρ*, *c_p_* and *λ* refer to density, specific heat and heat conductivity, respectively; *α* and *a* are volume fraction and specific surface area, referring to the ratios of the volume of a constituent or solid surface area to the whole volume of control cell, respectively; subscripts *t*, *o* and *w* stand for transformer, oil and winding, respectively; *h* is the convective heat transfer coefficient, calculated by the empirical correlation of Equation (3) [6]. The term *S_JH_* is the volumetric Joule heat and is calculated by *S_JH_ = I*^2^*R/α_t,w_* with *I* and *R* standing for current and electric resistance.
(3)$$Nu_{x}=\left\{\begin{array}{cc} \begin{array}{l} 1.49\cdot (x*{)}^{-1/3}\\ 1.49\cdot (x*{)}^{-1/3}-0.4\\ 8.235+8.68\cdot (10^{3}\cdot x*{)}^{-0.506}\cdot e^{-164x*} \end{array} & \begin{array}{l} \left(x*\leq 0.0002\right)\\ \left(0.0002<x*\leq 0.001\right)\\ \left(x*>0.001\right) \end{array} \end{array}\right.$$
Here, $Nu_{x}=\frac{h_{x}d}{\lambda }$ and x*=x/dRe·Pr with *d* for characteristic size.

The oil cooler is also expressed with the one-dimensional distributed parameters, i.e., the cross-sectional temperatures of oil and solid are assumed to be lumped at the centers, and their variations along the oil flow direction are identified. The heat capacities of oil and solid are much larger than that of air, and they can influence the dynamic temperature fields of the transformer. Thus, the heat transfer in the oil cooler is expressed by the following dual energy equations:(4)$$\rho_{o}c_{p,o}\frac{\partial T_{c,o}}{\partial t}+\rho_{o}c_{p,o}u_{c,o}\frac{\partial T_{c,o}}{\partial x}=\lambda_{o}\frac{\partial^{2}T_{c,o}}{\partial x^{2}}+h_{c,o}a_{c,o}\frac{\alpha_{c,s}}{\alpha_{c,o}}\cdot \Delta T_{so}$$
(5)$$\rho_{s}c_{p,s}\frac{\partial T_{c,s}}{\partial t}=\lambda_{s}\frac{\partial^{2}T_{c,s}}{\partial x^{2}}-h_{c,o}a_{c,o}\Delta T_{so}-h_{c,a}a_{c,a}\Delta T_{sa}$$
Here, the subscripts *c*, *a* and *o* stand for the oil cooler, air and oil, while subscript *s* for solid, including partitions and fins. The ∆*T_so_* and ∆*T_sa_* are the heat transfer temperature differences between the solid and oil or air, respectively. The convection heat transfer rates on the oil and air sides are enhanced with the serrated fins and wavy fins, whose empirical heat transfer correlations can be referred to as Refs. [16,17], respectively. For numerical solution, the oil cooler is discretized into many small heat exchangers in the oil flow direction. For each part, the ∆*T_so_* takes the arithmetic mean value, while the ∆*T_sa_* takes the logarithmic mean one, as shown in Equation (6) because the inlet and outlet temperatures differ greatly on the air side.
(6)$$\Delta T_{sa}=\frac{\left(T_{c,s}-T_{c,a}^{in})-(T_{c,s}-T_{c,a}^{out}\right)}{Ln[\left(T_{c,s}-T_{c,a}^{in}\right)/\left(T_{c,s}-T_{c,a}^{out}\right)]}$$
Here, the superscripts *in* and *out* refer to the values at the inlet and outlet of the discrete oil cooler element, respectively. The air outlet temperature is determined by the below thermal balance equation,
(7)$$T_{c,a}^{out}=T_{c,a}^{in}+Q/(q_{m,a}c_{p,a})$$
where *Q* and *q_m,a_* are the heat transfer rate and air flow rate of the discrete element, and the former is calculated by Newton’s cooling formula.

The oil pump runs with a flow rate varying with its lift head. Its actual oil flow rate depends on the balance between the lift of the oil pump and the total flow resistance of the transformer, oil cooler and pipes. Their flow resistances (△*p*) are all modeled by
(8)$$\Delta p=fL\rho u^{2}/\left(2d\right)$$ Here, *L* is the channel length; the *f* factor is related to Re number and for the channels with serrated fins it is calculated by the empirical correlation in Ref. [16].

The low- and high-voltage windings of current transformers take different geometries and working currents. To improve prediction accuracy, they are modeled separately and the oil is assumed to be mixed thoroughly at the outlet of the transformer.

Finally, with the assumption that the oil flow rate, temperature and pressure are continuous at the interfaces of adjacent components, the above submodels of transformer, oil cooler, pump and pipes are integrated into the heat dissipation model of ODAF traction transformers.

### 2.2. Numerical Solution

Finite difference method is used to convert the differential energy equations into algebraic ones, where the central difference scheme is adopted for the spatial derivatives and the fully implicit Euler scheme for time terms. The algebraic equations are solved with the sparse matrix method. The numerical solution performed on the Matlab platform takes the calculation flowchart depicted in Figure 2. It is involved in a number of switches between calculating the current temperature fields of the transformer and oil cooler and stepping into the next moment till the required time is met. As the windings take the electric resistance dependent on temperature, one iteration loop is adopted to correct the temperature fields for the heat generation rate. The synthetic ester dielectric fluid MIDEL 7131, acting as the transformer oil, takes the viscosity highly dependent on temperature, and thus an extra iteration loop is used to update the oil flow rate. When the maximal temperature variation between two adjacent moments is below the tolerance (1 × 10^−5^), steady heat dissipation is assumed to be obtained, then stop computation and export results.

Numerical computation is performed on a personal computer with an Intel CPU of Intel(R) Core(TM) i7 CPU. Solution independence on mesh size and time step are checked. It is found that 30 segments for the traction transformer and 1 s for the time step are enough to generate consistent temperature fields compared with those of 60 segments and 2 s. Thus, these parameters are applied for the final computation. The numerical computation is robust with the relaxation factor (γ) of 0.5, and it takes ~30 min to simulate 2 h’ running. Parallelization is not applied since the mesh number is limited.

## 3. Experimental Work and Model Validation

### 3.1. Temperature Rise Experiment

For the temperature rise experiment, a core-type of traction transformer is built. It has two coil columns with low-voltage winding wrapped near the core. The length of coil columns and the diameter of core limbs are about 730 and 200 mm, respectively. The high-voltage winding takes a rated voltage of 25 kV and a current of 147 A, while the low-voltage has a voltage of 950 V and a current of 4 × 966 A. In the circumferential direction, 18 spacers are arranged between adjacent winding layers, as depicted in the A-A view of Figure 1. The spacer thicknesses are 3.5 and 2.5 mm for the low- and high-voltage windings. Additionally, the oil pipes take a diameter of 90 mm, and the rated oil flow rate is about 50 m^3^/h.

The traction transformer is assembled with two oil coolers, an oil pump and pipes, as depicted in Figure 3a. The test rig is depicted in Figure 3b, comprising the transformer assembly and data monitoring and acquisition system. Thermocouples are set at the inlet and outlet of the transformer to monitor the oil temperature variations. The average temperatures of low- and high-voltage windings are obtained by analyzing the corresponding measurements of winding electric resistances. The windings and oil have an ambient temperature of 31 °C at the start of the experiment. After running for about an hour, their temperatures become independent of time. Continue running for another hour, then stop and export data.

### 3.2. Experimental Results and Model Validation

The above temperature rise experiment is simulated with the current traction transformer model. The numerical dynamic oil temperatures at the inlet and outlet of the transformer (with solid marks) together with experimental counterparts (with cross marks), are depicted in Figure 4a. It is seen that the numerical oil temperatures take the same variation trend as experimental counterparts, i.e., they increase sharply at the initial stage of powering on (for about 27 min), and then their increment rates decrease. Finally, the oil temperatures tend to be stable. These variation features can be related to the unsteady heat transfer in the transformer together with the secondary heat dissipation of hot oil in oil coolers, which is detailed below:

During the initial runtime, the oil has a large viscosity due to the low temperature; thus, the transformer, oil cooler and pipes have a large flow resistance and the oil pump runs with a small flowrate, which is not conducive for oil to absorb Joule heat from windings. On the other hand, under the condition that the oil has a low temperature and small flow rate, the heat dissipation in the oil cooler is weak and the oil heat absorption cannot be released to the ambient timely. The oil and winding temperatures are expected to rise sharply at the initial stage with the above two factors. However, after a period of operation, the oil temperature has a large rise, and its viscosity drops notably. Then, the heat transfer rates in the transformer and oil coolers are strengthened, and the heat generated in the windings can be discharged into the ambient more effectively due to the larger oil flow rate. As a result, the oil and winding temperatures increase at a decreasing rate and finally become stable.

Comparing the curves in Figure 4a quantitatively, it is seen that the numerical inlet and outlet oil temperatures of the transformer match the experimental counterparts well. Their deviations under stable operation conditions equal 1.9 and 0.7 °C, respectively, while the maximum deviation over the whole dynamic process is less than 5 °C. In addition, the running time for a stable operation, here defined as the time over which the outlet oil temperature reaches the value with 1 °C difference from the stable counterpart, matches well with the experimental value, i.e., 62 vs. 65 min.

The mean temperatures of low- and high-voltage windings under stable working conditions are measured in the experiment, and they are shown with cross marks in the upper right corner of Figure 4b. For the convenience of comparison, the temperature fields of low- and high-voltage windings calculated by the current model are averaged spatially, and their temporal variations are plotted in Figure 4b. From the curves in Figure 4b, it is seen that the winding temperatures rise rapidly at the initial operation stage. After a period of running, their increment rates drop notably, and the winding temperatures finally tend to be stable. These variation trends of winding temperatures are consistent with those of oil in Figure 4a and are reasonable because it is the oil that takes away the windings’ Joule heat. Quantitatively, the current predicted temperatures of low- and high-voltage windings agree well with experimental values in Figure 4b, and their relative deviations equal about 0.6% (92.7 vs. 92.1 °C) and 3.2% (81.4 vs. 78.9 °C), respectively.

From the above comparisons, it can be concluded that the current heat dissipation model of traction transformers has reasonable accuracy and can be used to further study the dynamic temperature field of traction transformers.

## 4. Discussions on Transformer Heat Dissipation

In order to understand in depth the dynamic heat transfer in traction transformers, Figure 5 presents with solid marks the axial variations of windings temperatures at different moments (i.e., t = 250, 500, 1000, 1500, 2000 and 3000 s) along with the corresponding oil counterparts with open marks. Since the oil absorbs the Joule heat of windings during the flow through narrow channels, one can expect that its temperature takes the rising feature in the oil flow direction. With the consideration that the convection heat transfer between windings and oil is driven by their temperature difference, the windings can also take the temperature increase in the axial direction. Scrutinizing the curves in Figure 5, it is seen that for all the moments, the windings and oil take the temperatures rising longitudinally, consistent with the above deductions. Additionally, the local oil and winding temperatures increase with time, and their temporal increment rates decrease, consistent with curves in Figure 4a,b.

In the current work, the electric resistance of copper wires and the Joule heat of windings increase with the increment of temperature linearly. However, the curves in Figure 5 show that the axial variation rate of windings temperature near the top of the transformer (x = 720 mm) is small. This spatial temperature profile can be related to the fact that the oil property is greatly dependent on temperature. In more detail, the current oil viscosity decreases notably with the rise of its temperature; thus, the convection heat transfer at the top of the transformer is much better than that at the bottom. Therefore, to transfer the same heat flux, a smaller temperature difference is required near the top of the transformer. On the other hand, as the current oil heat capacity (equal to the product of oil flow rate and specific heat) is large, its temperature increase is limited even after the absorption of a large amount of heat. As a result, although the downstream winding has a larger heat generation, its axial temperature increment rate may be smaller than the upstream counterpart. Carefully examining the temperature curves of oil and windings in Figure 5, it is seen that the axial increase rate of oil temperature is significantly smaller than that of the windings, and the local heat transfer temperature difference increases in the flow direction, which indicates the downstream winding transfers more heat than the upstream, consistent with the fact that downstream winding has a greater heat generation rate due to a higher temperature. The above-mentioned axial variation trend of winding temperature accords with that reported in [13].

The oil and winding temperatures become independent of time after a long run, and Figure 6 depicts their axial profiles. Scrutinizing the curves in Figure 6, it is seen that their temperatures rise in the oil flow direction. Additionally, despite the same oil temperature at the inlets, the low-voltage windings (with solid marks) have a higher temperature and a larger increment rate compared with the high-voltage counterparts, which can result from several factors. In more detail, compared with low-voltage winding, the high-voltage winding can have a larger volumetric heat generation rate due to the smaller single-layer thickness; on the other hand, it takes narrow gaps to enhance the convection heat transfer. Moreover, the non-uniform gaps can induce different flow rates in low- and high-voltage windings. Thus, to obtain uniform temperature, one needs properly specify the structural parameters for the low- and high-voltage windings.

Ambient temperature exerts a great effect on the winding temperatures of traction transformers by influencing the heat transfer rate of oil coolers. Figure 7 depicts the temperatures of low- and high-voltage windings under conditions with different ambient temperatures. It is seen that with the increment of ambient temperature, the winding temperatures at the bottom and top of the transformer, along with that of the discharged oil, increase. Additionally, the increment rates of winding and oil temperatures are found to exceed that of ambient temperature. In more detail, when the ambient temperature rises from 15 to 45 °C, the temperature increments of low- and high-voltage windings are equal to 37.1 and 36.0 °C, respectively, while the oil temperature increases by 34.7 °C. These phenomena are consistent with the fact that the Joule heat of winding rises with the increment of winding temperature.

## 5. Conclusions

In the present work, a novel mathematic model of heat dissipation along with its numerical solution algorithm is presented for the ODAF or “Oil Directed Air Forced” traction transformers, where winding and oil temperature fields are both expressed by distributed parameters. Additionally, the temperature rise experiment is performed with a traction transformer for model validation. Finally, the fundamentals of dynamic windings heat dissipation are discussed based on the conjugate heat transfer in the transformer and secondary heat release in the oil cooler, together with the oil flow rate of the pump. Research results demonstrate that the novel traction transformer model can simulate the temporal and spatial temperature variations of transformers at a small computation cost, and its numerical precision is reasonable with the relative deviations of winding temperature and dynamic transition time equal to ~2.5 °C and 3 min against experimental counterparts. Moreover, it is observed that the geometry parameters and ambient temperature could have a great influence on the temperature profiles of traction transformers.

The current work can be applied to study the effects of geometrical and operational parameters on heat dissipation and flow resistance performances, which is conducive to the thermal design and optimization of traction transformers. Additionally, a traction transformer often runs with its power and ambient temperature varying with time; thus, some regulations need to be imposed on its fan and oil pump to save energy. To solve this problem, authors are arranging some work for the development of the current model.

## Figures and Tables

**Figure 1 entropy-25-00457-f001:**
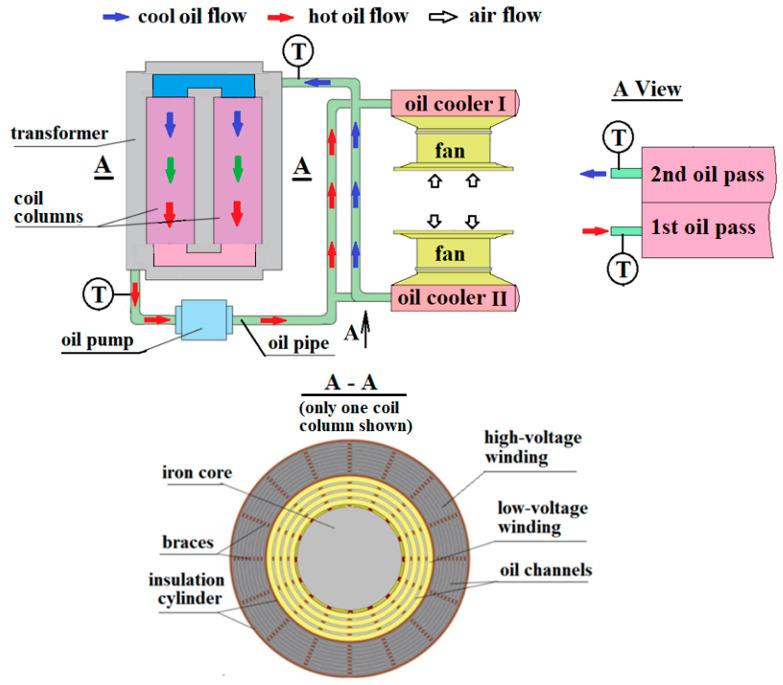
Schematic of heat dissipation of traction transformers with ODAF or “Oil Directed Air Forced” scheme, along with arrangement of low- and high-voltage windings in coil column.

**Figure 2 entropy-25-00457-f002:**
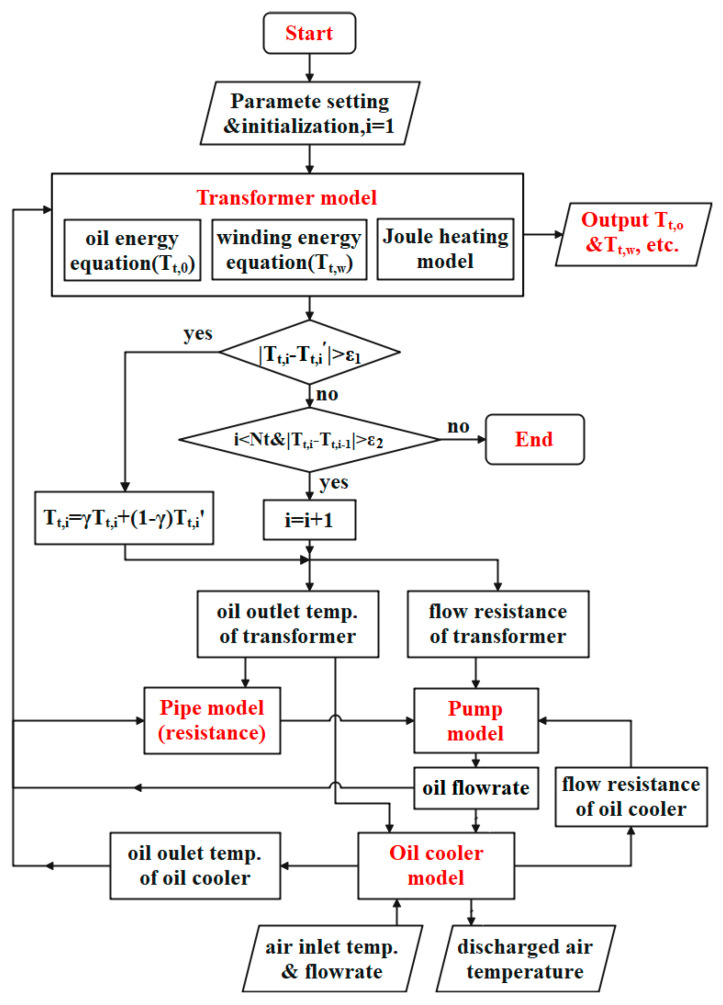
Calculation flowchart for heat dissipation of traction transformer with an oil-directed circulation.

**Figure 3 entropy-25-00457-f003:**
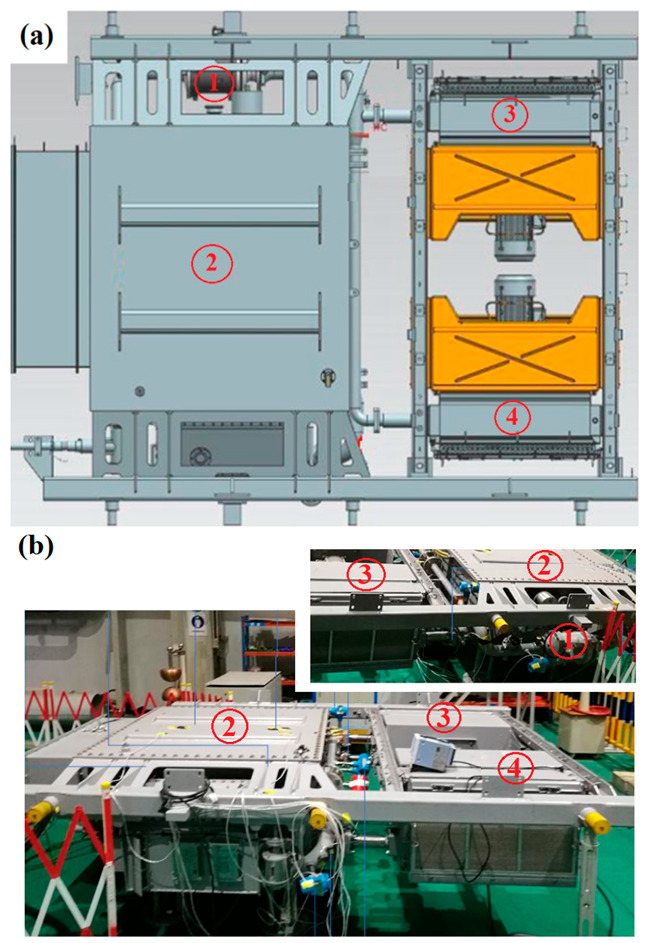
Schematics for (**a**) assembly of traction transformer and its cooling system, and (**b**) test rig for temperature rise experiment, where 1 stands for pump, 2 for traction transformer; 3 for oil cooler I and 4 for oil cooler II.

**Figure 4 entropy-25-00457-f004:**
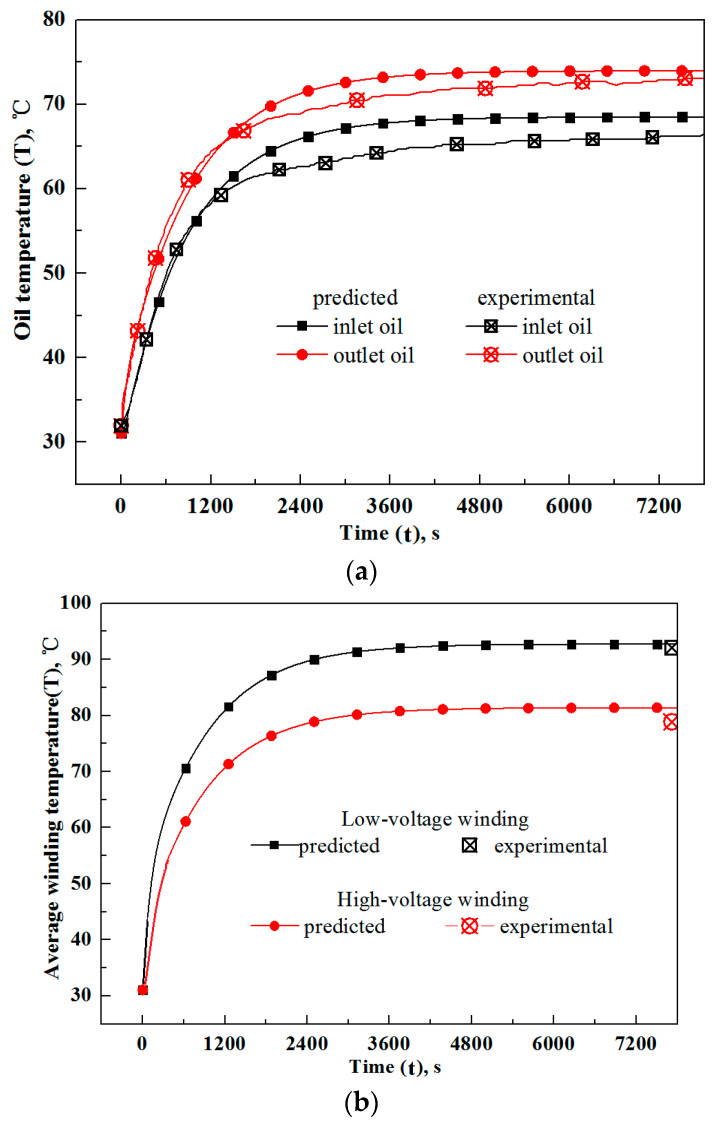
Comparisons of dynamic transformer temperatures obtained by current model with those of experimental work. (**a**) Oil temperatures; (**b**) Average winding temperatures.

**Figure 5 entropy-25-00457-f005:**
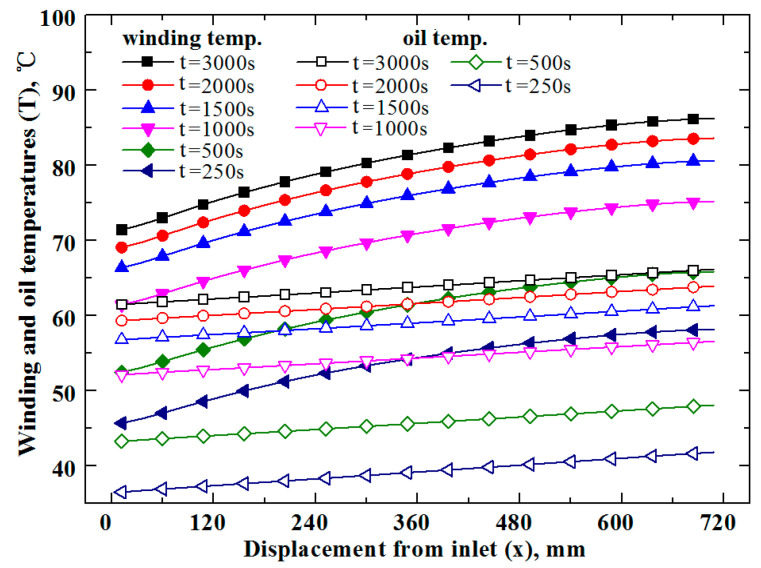
Variations of windings and oil temperatures with time and space in traction transformer.

**Figure 6 entropy-25-00457-f006:**
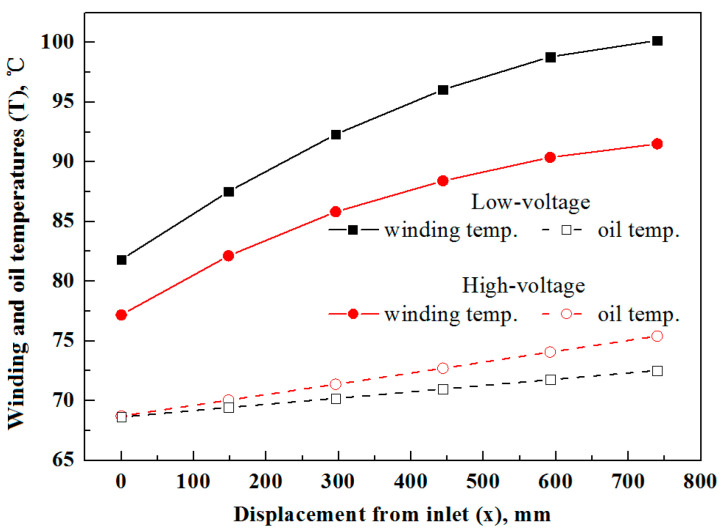
Axial temperature profiles of low- and high-voltage windings and oil under stable working conditions.

**Figure 7 entropy-25-00457-f007:**
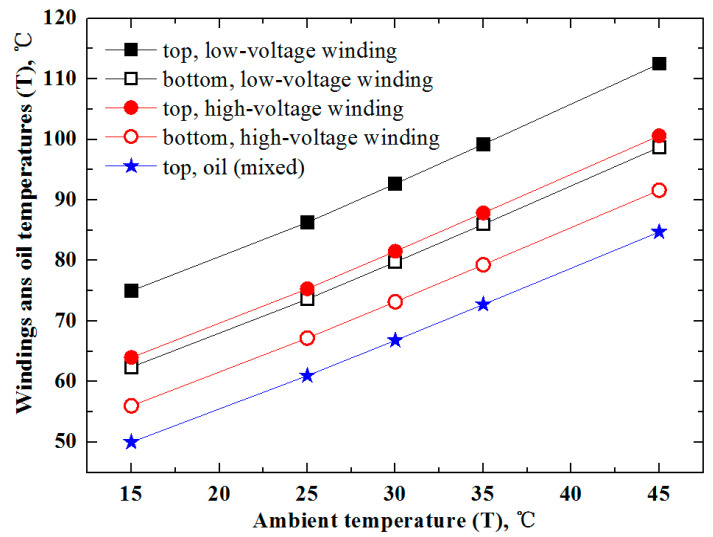
Effects of ambient temperature on winding and oil temperatures in traction transformer.

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
