# Peer review of "Dynamic Heat Dissipation Model of Distributed Parameters for Oil-Directed and Air-Forced Traction Transformers and Its Experimental Validation"

_entropy, 2023, doi:10.3390/e25030457_

Round 1
Reviewer 1 Report
The article is of good quality and correctly written as required. It is properly divided into chapters. The quality of the drawings is good and they are legible. The drawings are numbered and correctly described. Dependencies are legible and numbered. The article contains a list of designations and a list of references. The article is of good quality and can be published in this form.
Author Response
The authors are grateful for the reviewer‘s positive comments.

Reviewer 2 Report
The manuscript, entitled "Dynamic Heat Dissipation Model of Distributed Parameters for Oil-Directed and Air-Forced Traction Transformers and Its Experimental Validation", effectively integrates both theoretical and experimental approaches in the study of oil-directed and air-forced traction transformers. The authors have made commendable efforts in clearly presenting the data and parameters through the use of a comprehensive nomenclature, making the information accessible to readers who may not be familiar with the mathematical formalisms employed.
However, my primary concern lies with the conclusion, which fails to explicitly highlight the limitations and strengths of the proposed models in describing the transformers studied.
Author Response
Authors are grateful for the positive comments from the reviewer. To describe the limitations and strengths of the proposed models, a new paragraph, presented as below, is added in the conclusion section of the revised manuscript.
The current work can be applied to study the effects of geometrical and operational parameters on the heat dissipation and flow resistance performances, which is conducive for the thermal design and optimization of traction transformers. Besides, a traction transformer often runs with its power and ambient temperature varying with time, thus some regulations need be imposed on its fan and oil pump to save energy. To solve this problem, authors are arranging some work for the development of current model.
